# The Mitochondrial PHB2/OMA1/DELE1 Pathway Cooperates with Endoplasmic Reticulum Stress to Facilitate the Response to Chemotherapeutics in Ovarian Cancer

**DOI:** 10.3390/ijms23031320

**Published:** 2022-01-25

**Authors:** Meiyu Cheng, Huimei Yu, Qinghuan Kong, Bingrong Wang, Luyan Shen, Delu Dong, Liankun Sun

**Affiliations:** Department of Pathophysiology, College of Basic Medical Sciences, Jilin University, Changchun 130021, China; mycheng19@mails.jlu.edu.cn (M.C.); yuhuimei@jlu.edu.cn (H.Y.); akashihime@163.com (Q.K.); wangbr19@mails.jlu.edu.cn (B.W.); shenly@jlu.edu.cn (L.S.)

**Keywords:** OMA1, ovarian cancer, DELE1, mitochondrial membranes, endoplasmic reticulum stress

## Abstract

Interactions between the mitochondrial inner and outer membranes and between mitochondria and other organelles closely correlates with the sensitivity of ovarian cancer to cisplatin and other chemotherapeutic drugs. However, the underlying mechanism remains unclear. Recently, the mitochondrial protease OMA1, which regulates internal and external signals in mitochondria by cleaving mitochondrial proteins, was shown to be related to tumor progression. Therefore, we evaluated the effect of OMA1 on the response to chemotherapeutics in ovarian cancer cells and the mouse subcutaneous tumor model. We found that OMA1 activation increased ovarian cancer sensitivity to cisplatin in vivo and in vitro. Mechanistically, in ovarian cancer, OMA1 cleaved optic atrophy 1 (OPA1), leading to mitochondrial inner membrane cristae remodeling. Simultaneously, OMA1 induced DELE1 cleavage and its cytoplasmic interaction with EIF2AK1. We also demonstrated that EIF2AK1 cooperated with the ER stress sensor EIF2AK3 to amplify the EIF2S1/ATF4 signal, resulting in the rupture of the mitochondrial outer membrane. Knockdown of OMA1 attenuated these activities and reversed apoptosis. Additionally, we found that OMA1 protease activity was regulated by the prohibitin 2 (PHB2)/stomatin-like protein 2 (STOML2) complex. Collectively, OMA1 coordinates the mitochondrial inner and outer membranes to induce ovarian cancer cell death. Thus, activating OMA1 may be a novel treatment strategy for ovarian cancer.

## 1. Introduction

Frequent cisplatin resistance seriously limits the treatment of ovarian cancer. Ovarian cancer cells exposed to cisplatin often change the tumor microenvironment [1], enhance metastatic properties [2], and adjust signal pathways [3], and consequently, they become resistant to drugs. Therefore, a multidisciplinary approach to explore the molecular mechanism of improving the efficacy of cisplatin is needed [4]. Recent studies have shown that, compared to the mitochondrial outer membrane (MOM), instability of the mitochondrial inner membrane (MIM) not only activates a variety of proteases that cleave mitochondrial pro-apoptotic proteins such as Diablo, phosphoglycerate mutase 5 (PGAM5), and HTRA2 [5,6] but also affects MOM permeability through membrane shaping proteins such as S-OPA1 [7]. Coordination of the MIM and MOM is thought to be the premise for mitochondrial-mediated apoptosis [8]. Therefore, exploring the coordination of MIM and MOM may provide new clues that could increase responses to chemotherapeutics such as cisplatin in ovarian cancer.

Dynamic MIM responses are triggered by drug-induced stress. OMA1 is an integral MIM protease [9] that is highly expressed in colorectal cancer and gastric cancer [10,11]. OMA1 may promote tumor development by driving metabolic reprogramming and increasing oxidative stress [10,11]. However, another study found that OMA1 reduced the MIM cristae by degrading optic atrophy 1 (OPA1) in a murine colorectal tumor model and in HCT116 xenograft tumors [10]. In human embryonic kidney (HEK293T) cells and mouse embryonic fibroblast cells (MEFs), OMA1 was shown to maintain the MIM–MOM connectivity by interacting with mitochondrial contact site and cristae junction organizing system (MICOS), promoting apoptosis [12]. These findings suggest that OMA1 is involved in the coordination of MIM and MOM. Nevertheless, a role for OMA1 in ovarian cancer has not been established. Understanding the role of OMA1 in ovarian cancer may help to explain the mechanism of anticancer agents on mitochondrial-mediated apoptosis.

Communication between organelles may be one of the main factors driving mitochondrial outer membrane permeability (MOMP). Currently, MOMP is thought to be primarily regulated by the BCL2 family proteins [13]. Our previous results showed that in ovarian cancer cisplatin-resistant cells (SKOV3/DDP), inhibiting BCL2 increased cisplatin-induced contact between mitochondria and the endoplasmic reticulum (ER), which initiated the mitochondrial apoptotic pathway [14], suggesting that the interaction between the mitochondria and the ER is related to MOMP and cisplatin sensitivity. Another study showed that instability of ER mitochondrial-associated membranes (MAMs) reduced the distribution of BAX on MOM, thus reducing MOMP [15]. It has been further suggested that, in addition to this direct effect, the communication between ER and mitochondria may be related to a cooperative mechanism between the MIM and MOM.

Recently, OMA1 was reported to cleave the mitochondrial protein DELE1, facilitating its cytoplasmic release [16,17]. In the cytoplasm, DELE1 combines with EIF2AK1 to activate EIF2S1 [16]. EIF2AK1 and the ER stress sensor EIF2AK3 have extensive homology in their kinase catalytic domains and both belong to EIF2S1 kinases [18]. EIF2AK1 and EIF2AK3 have overlapping functions and can cooperate to specifically regulate cellular responses to a variety of stressors [19]. Therefore, we hypothesized that EIF2AK1 may cooperate with EIF2AK3 to amplify interactions between mitochondria and the ER, which could increase MOMP and chemosensitivity in ovarian cancer.

The membrane scaffold proteins prohibitin 2 (PHB2) and stomatin-like protein 2 (STOML2) belong to the stomatin, prohibitin, flotillin, HflC/K (SPFH) family. They form a membrane domain with a specific lipid and protein composition in cells and regulate the turnover of membrane proteins [20,21]. In mitochondria, PHB2 and STOML2 are anchored to the MIM to regulate mitochondrial protease activity [21]. PHB2 and STOML2 are reportedly overexpressed in ovarian cancer, which is related to mitochondrial function and apoptosis resistance, but the mechanism remains unclear [22,23]. In neurons, the PHB1/2 complex regulates turnover of the mitochondrial protease OMA1 by stabilizing cardiolipin [24]. STOML2 was also found to interact with exogenous OMA1 in FITR393T cells [25]. PHB2 often forms a complex with STOML2 to maintain its stability [26], so the PHB2/STOML2 complex is likely to participate in OMA1 regulation. Exploring the regulation of OMA1 by the PHB2/STOML2 complex will help to illustrate a potential mechanism of targeting mitochondria to increase the chemosensitivity of ovarian cancer cells.

## 2. Results

### 2.1. FCCP Enhanced Cisplatin-Induced Apoptosis in Ovarian Cancer Cells

To explore the role of mitochondria in the chemosensitivity of ovarian cancer, the human ovarian cancer cells A2780 and mouse ovarian cancer cells ID8 were treated with the mitochondrial uncoupler carbonyl cyanide p-trifluoromethoxyphenylhydrazone (FCCP) and/or cisplatin. Cell viability was slightly inhibited after FCCP or cisplatin treatment but was significantly decreased after combined treatment with FCCP and cisplatin, as assessed from the MTT assay (Figure 1a,b). Next, we investigated whether FCCP combined with cisplatin could induce apoptosis. Annexin V/PI staining revealed that the rate of apoptosis increased slightly after FCCP or cisplatin treatment but increased when the treatment was combined (Figure 1c–f). Western blotting results revealed that expression of the apoptotic proteins, cleaved caspase-3 and cleaved caspase-9, was increased after combined treatment compared to that in the control (Figure 1g,h). These data indicated that FCCP enhanced cisplatin-induced apoptosis in A2780 and ID8 cells.

### 2.2. FCCP Combined with Cisplatin Activated OMA1 and Induced Optic Atrophy 1 (OPA1) Cleavage and Mitochondrial Fragmentation

FCCP, a mitochondrial uncoupler, activates mitochondrial proteases and affects mitochondrial homeostasis [8,21]. We, therefore, investigated the expression of the mitochondrial proteases, OMA1 and PARL, in both sets of cells. The results showed that, after treatment with FCCP and/or cisplatin for 6 and 12 h, OMA1 was activated, the expression of PARL was reduced, and OPA1 and PGAM5 were cleaved in the group treated with FCCP alone and in that receiving the combined treatment (Figure 2a–d). Mitochondria are in a dynamic balance of continuous fusion and fission. This led us to investigate mitochondrial morphology via MitoTracker Red staining in A2780 cells. The results showed that the mitochondria in the control group were filamentous and connected into a network, and those in the group treated with cisplatin alone were still filamentous, whereas those in the group treated with FCCP alone were in the intermediate transition state between filamentous and punctate. Punctate accumulation was evident in mitochondria of the combined treatment group (Figure 2e), suggesting that these mitochondria were excessively fragmented, and the cells were dying.

### 2.3. FCCP Combined with Cisplatin Induced Mitochondrial Inner Membrane (MIM) Cristae Remodeling and Cytochrome c Release 

To further address the mechanism of apoptosis induced by the combined treatment with FCCP and cisplatin, we examined the morphology of mitochondrial cristae in A2780 cells. The results showed that the mitochondrial cristae in the control group and the group treated with cisplatin alone were normally shaped, while those in the group treated with FCCP alone and the group treated with both FCCP and cisplatin, decreased, loosened, or even disappeared (Figure 3a). Next, we isolated mitochondria and cytoplasm and found that, compared to that in the control group, the cytochrome c in the cytoplasm of the group treated with cisplatin alone and the group treated with FCCP alone did not change significantly, while the content of cytochrome c in the cytoplasm of the combined treatment group increased (Figure 3b,c). Immunofluorescence results showed that cytochrome c was released into the cytoplasm only in the combined treatment group (Appendix A). Next, the mitochondrial membrane potential and the integrity of MOM were explored. Results showed that the mitochondrial membrane potential decreased in the group treated with FCCP alone and in the combined treatment group (Figure 3d,e); however, the MOM was still intact in the group treated with FCCP alone, whilst the MOM was ruptured in the combined treatment group (Figure 3a). These data suggest that the remodeling of MIM cristae and the rupture of MOM are both required for the release of cytochrome c, and that the mitochondrial membrane potential is not enough to represent the permeability of MOM.

### 2.4. Activated OMA1 Cooperated with EIF2AK3 to Activate the EIF2S1/ATF4 Pathway by Cleaving DELE1

In response to stress signals, mitochondria release a variety of substances, which interact with cytoplasmic proteins and induce apoptosis [5,6,16,17]; therefore, we detected DELE1 in the cytoplasm. Owing to the lack of an effective antibody against DELE1 [17], we constructed a DELE1-HA overexpression plasmid. The results showed that DELE1 was released into the cytoplasm (Figure 4b) and interacted with EIF2AK1 in the group treated with FCCP alone and in the combined treatment group (Figure 4c,d). EIF2AK1 is an EIF2S1 kinase that mediates EIF2S1 phosphorylation on serine 51 [27]. The phosphorylation level of EIF2S1 was assessed in both sets of cells; the result showed that EIF2S1 phosphorylation slightly increased in the group treated with FCCP alone but was significantly increased in the combined treatment group (Figure 4e,f), suggesting that EIF2S1 phosphorylation might be regulated by other phosphatases. Considering that EIF2S1 is also a key protein in regulating ER stress, we examined the ER stress sensor EIF2AK3 [27] and its phosphorylation and found that the expression of p-EIF2AK3 increased in the group treated with cisplatin alone and in the combined treatment group (Figure 4e,f). These results suggested that cisplatin and FCCP activated EIF2AK3 and EIF2AK1, respectively, and jointly enhanced EIF2S1 phosphorylation. The transcription level of *ATF4*, a downstream molecule of EIF2S1, and its expression in the nucleus were found to increase (Figure 4g,i), indicating that EIF2S1 promoted the transposition of ATF4 into the nucleus. Since we had determined that apoptosis had occurred, we detected the death-related molecules, ATF5 and DDIT3, which are present downstream of ATF4. qRT-PCR results showed that the transcription levels of *ATF5* and *DDIT3* were markedly upregulated in the combined treatment group (Figure 4i), and the expression of ATF5 and DDIT3 in the nucleus was increased (Figure 4g,h), indicating that ATF4 promoted the transcription and nuclear transfer of ATF5 and DDIT3. As the transcription factors DDIT3 and ATF5 primarily regulate the BCL2 family proteins [28,29], transcriptional levels of the BCL2 family proteins in A2780 cells were examined; results showed that the anti-apoptotic protein MCL1 was downregulated, while the pro-apoptotic proteins BCL2L11, PMAIP1, and BBC3 were upregulated (Figure 4j). Mitochondria were isolated, and the distribution of BAX in the mitochondria was found to have increased (Figure 4k,l), which explained the rupturing of MOM (Figure 3a).

### 2.5. Knockdown of OMA1 Partially Reversed Apoptosis by Reducing the Mitochondrial Stress and EIF2S1/ATF4 Pathway

To further investigate the function of OMA1 in apoptosis, OMA1 in A2780 cells was knocked down using shRNA. FCCP and cisplatin were added to observe whether knockdown of OMA1 could reverse apoptosis. Annexin V/PI staining showed that knockdown of OMA1 partially rescued the apoptosis induced by the combined treatment with FCCP and cisplatin (Figure 5a,b). MTT assay results showed that knockdown of OMA1 partially rescued the inhibitory effect on cell viability induced by the combined treatment with FCCP and cisplatin (Figure 5c). These data suggested that knockdown of OMA1 partially reversed the apoptosis induced by FCCP when combined with cisplatin.

Next, the molecular mechanism underlying apoptosis reversal of OMA1 was explored. Western blotting results revealed that knockdown of OMA1 attenuated the cleavage of OPA1 and DELE1 induced by the combined treatment with FCCP and cisplatin (Figure 5d,f). Simultaneously, the mitochondrial morphology was found to be in the intermediate state between filamentous and punctate (Figure 5e), and the expression of p-EIF2S1, ATF4, ATF5, and DDIT3 was decreased (Figure 5g). Next, we isolated mitochondria from the cytoplasm and found that the content of cytochrome c in the cytoplasm of the FCCP + cisplatin + OMA1-shRNA group was reduced compared to that of the FCCP + cisplatin + Src-shRNA group (Figure 5h). These data suggested that knockdown of OMA1 reversed the apoptosis induced by the combined treatment with FCCP and cisplatin through attenuating the mitochondrial stress and the EIF2S1/ATF4 pathway. 

### 2.6. The OMA1 Protease Activity Was Regulated by the Prohibitin 2 (PHB2)/Stomatin-like Protein 2 (STOML2) Complex 

The expression of PHB2 and STOML2 in A2780 cells was investigated, and it was found that the total expression of PHB2 and STOML2 did not change significantly after treatment with FCCP and/or cisplatin for 12 h (Figure 6a). Furthermore, the expression of STOML2 in mitochondria was slightly increased in the combined treatment group. Compared with that in the control group, PHB2 expression in the mitochondria increased in the group treated with FCCP alone or cisplatin alone and decreased in the combined treatment group (Figure 6b). The interactions between STOML2, PHB2, and OMA1 in both cells, when investigated via immunoprecipitation and revealed that STOML2/PHB2 and STOML2/OMA1 binding decreased in the group treated with FCCP alone and in the combined treatment group (Figure 6c–f), suggesting that the destruction of the STOML2/PHB2 complex activated OMA1.

### 2.7. Effects of FCCP and Cisplatin Treatments in a Mouse Subcutaneous Tumor Model 

To analyze the effect of FCCP and cisplatin in vivo, we established a mouse subcutaneous tumor model. The results indicated a slight inhibition of tumor growth after FCCP or cisplatin treatment and significant inhibition of tumor growth in the combined treatment group (Figure 7a–d). TUNEL staining revealed that the combination of FCCP and cisplatin significantly increased cell apoptosis (Figure 7e). Simultaneously, OMA1 was activated and OPA1 was cleaved in the group treated with FCCP alone and the group treated with the combination of FCCP and cisplatin (Figure 7f). Consistent with the in vitro experiments, the expression of p-EIF2S1, p-EIF2AK3, ATF4, ATF5, and DDIT3 (Figure 7g); the expression of ATF4, ATF5, and DDIT3 in the nucleus (Figure 7h); and the transcription levels of *Atf4*, *Atf5*, and *Ddit3* (Figure 7i) increased in the combined treatment group. Additionally, the transcription of the anti-apoptotic protein MCL1 was downregulated, while that of the pro-apoptotic proteins BCL2L11, PMAIP1, and BBC3 was upregulated in the combined treatment group (Figure 7j). Simultaneously, the expression of the anti-apoptotic protein MCL1 decreased, while that of the pro-apoptotic proteins BAX, cleaved-caspase9, and cleaved-caspase3 increased in the combined treatment group (Figure 7k). Immunofluorescence studies showed that the co-localization of STOML2 with PHB2 decreased in the group treated with FCCP alone and in the combined treatment group (Figure 7l). These data indicated that FCCP combined with cisplatin inhibited tumor growth through inducing mitochondrial and ER stress via OMA1.

## 3. Discussion

We detected two forms of OMA1, in vivo and in vitro: L-OMA1 (60 KDa) and S-OMA1 (40 KDa). In the process of ovarian cancer cell apoptosis, L-OMA1 (60 KDa) accumulated and S-OMA1 (40 KDa) disappeared, indicating that OMA1 activation promoted apoptosis. This is consistent with the results and theories of Jiang et al. [8], who considered L-OMA1 (60 KDa) as an OMA1 precursor protein and S-OMA1 (40 KDa) as active OMA1. Once OMA1 was activated, it underwent autocatalysis and S-OMA1 (40 KDa) disappeared [8,30]. However, another study showed that the expression of S-OMA1 (40 KDa) increased during apoptosis [31], which was contrary to our results. This may be due to the differences in antibodies or cell lines used. The specific functions of the two forms of OMA1 need to be investigated further. Nonetheless, we determined that OMA1 was activated during this process, as the cleavage of OPA1 and DELE1 was attenuated after OMA1 knockdown. Although OMA1 is not the primary cause of ovarian cancer, its activation is involved in the apoptosis of ovarian cancer cells. A2780 are human endometrioid ovarian adenocarcinoma cells and ID8 are mice epithelial ovarian cancer cells, so our results cannot be a priori pertained to all subtypes of ovarian cancer.

OPA1 is a key regulatory protein for mitochondrial dynamics and mitochondrial cristae remodeling [32,33]. It exists in two forms: L-OPA1 and S-OPA1. Its cleavage is governed by proteases such as PARL and OMA1 [34]. Our results showed that OMA1 was activated, and PARL expression decreased during mitochondrial apoptotic cristae remodeling, indicating that the cleavage of L-OPA1 mediated by OMA1 may primarily mediate cell damage, while cleavage of L-OPA1 mediated by PARL may occur under normal conditions to maintain a low proportion of S-OPA1. Additionally, we found that the mitochondrial protein PGAM5 was also cleaved when OMA1 was activated and PARL expression decreased. This suggests that OMA1 partially replaces PARL to mediate the cleavage of PGAM5 when the function of PARL is attenuated. This was consistent with the results of Wai et al. [25], who found that a part of PGAM5 was still cleaved in PARL knockout cells, while depletion of OMA1 from PARL knockout cells completely prevented the cleavage of PGAM5. This implied that OMA1 is an integral mitochondrial protease.

Our experiments showed that the destruction of MIM cristae was not enough to initiate cell death, but rather that it primed the mitochondria for apoptosis. The cooperation of MOM was also required for cellular death. As the mitochondrial cristae were destroyed in the group treated with FCCP alone, the MOM was not ruptured and cytochrome c was not released. When FCCP was added alone, DELE1 was also released into the cytoplasm but with no rupture of MOM, indicating that the cytoplasmic release of DELE1 did not depend on the rupture of MOM. It was likely that some proteins mediated the transfer of DELE1 to the cytoplasm.

Our findings showed that the interaction between the mitochondria and ER was required for the rupture of MOM. Firstly, OMA1 cleaved DELE1, facilitating its cytoplasmic release, following which DELE1 interacted with and activated EIF2AK1. The activated EIF2AK1 cooperated with EIF2AK3 to induce EIF2S1 phosphorylation, and EIF2S1 phosphorylation reportedly induced the transcription of *DDIT3* and *ATF5*, which have pro-apoptotic functions [29]. Consistent with this result, we found that, both in vivo and in vitro, EIF2S1 phosphorylation promoted the entry of ATF4 into the nucleus, which upregulated DDIT3 and ATF5 expression to promote transcription of the pro-apoptotic proteins BBC3, BCL2L11, and PMAIP1 and to downregulate the transcription of the anti-apoptotic protein MCL1. The accumulation of BAX in the mitochondria and the rupture of the MOM were observed. A recent study showed that ATF4 directly bound the promoter region of *PMAIP1* to induce apoptosis of HeLa cells [35]. These findings suggested that, when stimulated by drugs, the traditional protective EIF2S1/ATF4 signaling pathway might be transformed into one of the pathways that induces cancer cell death.

Although OMA1 is activated by cellular stress, including the increase in ROS levels and the dissipation of mitochondrial membrane potential [30], the specific upstream molecules that regulate OMA1 remain largely unknown. Our results showed that the PHB2/STOML2 complex might regulate the activity of OMA1. The PHBs complex reportedly inhibits OMA1 activity, and the destruction of this complex releases OMA1 activity [24]. Similar to this result, we found that OMA1 was activated when STOML2/PHB2 binding andSTOML2/OMA1 binding decreased. Although we detected the PHB2/STOML2/OMA1 complex, we could not rule out the possibility that this complex contains other mitochondrial proteases as well. Exploring other proteases that bind to the PHB2/STOML2 complex will provide more in-depth mechanistic insights into the roles of PHB2 and STOML2 in mitochondrial homeostasis and chemosensitivity of ovarian cancer. Additionally, in the process of A2780 cell apoptosis, STOML2 expression in mitochondria increased and that of PHB2 decreased, indicating that the sublocalization of PHB2 and STOML2 in the cells was closely associated with their function. In the nucleus, PHB2 interacts with transcription factors, myogenic regulatory factors, and nuclear receptors to co-regulate transcription [26,36]. In the cytoplasm, PHB2 interacts with cell membrane proteins and receptors [26,36]. Therefore, exploring the relationship between the sublocalization of PHB2 and STOML2 and their functions will help to further illustrate their roles in the chemosensitivity of ovarian cancer.

Collectively, our results showed that the mitochondrial PHB2/OMA1/DELE1 pathway cooperated with cisplatin-induced ER stress to amplify the pro-death signals of MIM and MOM, which provided a novel mechanism for increasing the sensitivity of ovarian cancer to cisplatin (Figure 8). This study provides new insights into the role of OMA1 in the coordination of mitochondrial inner and outer membranes in ovarian cancer and indicates that OMA1 could be a potential target for the treatment of ovarian cancer.

## 4. Materials and Methods

### 4.1. Cell Lines and Cell Culture

A2780 cells and ID8 cells were purchased from the Chinese Academy of Medical Sciences and BeNa Culture Collection, respectively. A2780 cells were cultured in RPMI-1640 (Gibco Life Technologies, Carlsbad, CA, USA) and ID8 cells were cultured in high-glucose DMEM (Gibco Life Technologies). Both mediums were supplemented with 10% fetal bovine serum (Invitrogen, Carlsbad, CA, USA) at 37 °C in a 5% CO_2_. The OMA1 stable knockdown cell line was generated through infection of A2780 with OMA1-shRNA plasmid and selection with 800 μg/mL G418. 

### 4.2. Reagents and Antibodies

FCCP was obtained from Selleck (Houston, TX, USA); cisplatin was purchased from MCE (Monmouth Junction, NJ, USA); MTT was purchased from Sigma-Aldrich (St. Louis, MO, USA); PHB2 (12295-1-AP), SLP2 (60052-1-Ig), OMA1 (17116-1-AP), OPA1 (27733-1-AP), PARL (sc-514836), cytochrome c (10993-1-AP), ATF4 (10835-1-AP), ATF5 (sc-377168), CHOP (15204-1-AP), PERK (24390-1-AP), PGAM5 (28445-1-AP), and Actin (66009-1-Ig) antibodies were purchased from Proteintech ( Chicago, IL, USA); p-elF2α (3398S) antibody was purchased from Cell Signaling Technology (Danvers, MA, USA); and p-PERK (12814) antibody was purchased from Signalway Technology (St. Louis, MO, USA). All antibodies were used at a 1:1 K dilution.

### 4.3. Plasmids and Transfection

The pcDNA3.1 vector, pcDNA3.1-DELE1-HA and shRNA targeting human OMA1 were constructed by Sangon Biotech. The OMA1 shRNA sequences used were as follows: OMA1-shRNA 1: 5′-GGGCTGTCATCAAGTACAAGT-3′; OMA1-shRNA 2: 5′-GGAAGCTATTCCTTGGTTTGA-3′; OMA1-shRNA 3: 5′-GCTGTTAAAGAAGTGCTTTGT-3′. A2780 cells (6 × 10^5^ cells per well) were seeded into six-well plates and transfected with ViaFect^TM^ transfection reagent (Promega, Madison, MI, USA) following the manufacturer’s protocol.

### 4.4. Cell Viability Assay

A2780 cells (8 × 10^3^ cells per well) and ID8 cells (5 × 10^3^ cells per well) were seeded into 96-well plates and treated with FCCP and/or cisplatin for 24 h. MTT was added, and cells were incubated for 4 h. Media were removed, and 150 μL DMSO was added to dissolve formazan crystals. The absorbance at 570 nm was measured using a Multiskan Spectrum (BioTek, Winooski, VT, USA). 

### 4.5. Flow Cytometry 

A2780 cells (4 × 10^5^ cells per well) and ID8 cells (2 × 10^5^ cells per well) were seeded into six-well plates and treated with FCCP and/or cisplatin. Cells were harvested and stained according to the manufacturer’s instructions. Apoptosis was evaluated using an Annexin V-FITC PI double staining kit (BD Biosciences, San Diego, CA, USA). The mitochondrial membrane potential (MMP) was measured using the Mitochondrial Membrane Potential Assay Kit with JC-1 (Beyotime Biotechnology, Shanghai, China). Samples were analyzed by Guava easyCyte flow cytometry (Guava easyCyte, Austin, TX, USA).

### 4.6. Western Blotting 

A2780 cells (6 × 10^5^ cells per well) and ID8 cells (3 × 10^5^ cells per well) were seeded into six-well plates. Cell lysates were mixed with 5 × SDS-PAGE Protein Loading Buffer (Yeason, Shanghai, China) and boiled at 95 °C. Protein samples were loaded onto PAGE gels and transferred to PVDF. After blocking with 5% non-fat milk dissolving in PBST for 1.5 h at RT, membranes were incubated with primary antibody overnight at 4 °C. The next day, membranes were washed and incubated with horse radish peroxidase (HRP)–conjugated secondary antibodies for 1.5 h at RT. Bands were detected using Syngene Bio Imaging (Synoptics, Cambridge, UK). 

### 4.7. Immunofluorescence and Fluorescence Microscopy

A2780 cells (5 × 10^4^ cells per well) were seeded into 24-well plates and treated with FCCP and/or cisplatin. Cells were fixed in 4% paraformaldehyde for 15 min and permeabilized with 0.1% Triton X-100 for 7 min. Cells were blocked with 5% bovine serum albumin at RT for 30 min and subsequently incubated with primary antibody overnight at 4 °C. Cells were incubated, at RT, with FITC/Texas Red–conjugated secondary antibodies (Proteintech, Chicago, IL, USA) for 0.5 h and with Hoechst 33342 for 5 min. The images were acquired using an Echo Lab Revolve microscope (San Diego, CA, USA).

### 4.8. Transmission Electron Microscopy 

A2780 cells (1.2 × 10^6^ cells per dish) were seeded into 10 cm dishes and treated with FCCP and/or cisplatin. Cells were harvested and fixed in 2.5% glutaraldehyde overnight. Samples were washed in PB buffer and then postfixed with 1% osmium tetroxide for 2 h at 4 °C. After washing in double distilled water, samples were dehydrated with graded series of ethanol (50%, 70%, 90%, and 100%), and finally embedded in epoxy resins. Ultrathin sections (70 nm) were stained with 3% uranium acetate–lead citrate and imaged with a HITACHI 7800 TEM ( HITACHI, Tokyo, Japan). 

### 4.9. RNA Extraction and qRT-PCR 

A2780 cells (6 × 10^5^ cells per well) were seeded into six-well plates and treated with FCCP and/or cisplatin for 6 h. RNA extraction and qRT-PCR were performed as described before [37]. Standard cycling conditions (94 °C, 5 s; 55 °C, 15 s; 72 °C, 10 s; 40 cycles) were used. Primers sequences are presented in Table 1. The relative expression levels between groups were calculated by the ΔΔCt method [38] and normalized to the expression of *ACTB*.

### 4.10. Co-Immunoprecipitation

A2780 cells (1.2 × 10^6^ cells per dish) and ID8 cells (6 × 10^5^ cells per dish) were seeded into 10 cm dishes and treated with FCCP and/or cisplatin. Cells were lysed in NP-40 buffer with 1× protease inhibitor. Lysates were shaken slowly at 4 °C for 45 min and centrifugated at 14,000× *g* for 15 min at 4 °C. The supernatants were incubated with 2 μg primary antibody and shaken slowly at 4 °C overnight. Then, 30 μL protein G agarose (Beyotime Biotechnology, Shanghai, China) was added and shaken slowly at 4 °C for 3 h, followed by centrifugation at 2500 rpm for 5 min. Pellets were washed three times with PBS. After the last centrifugation, 5 × SDS-PAGE Protein Loading Buffer and RIPA buffer were added to resuspend pellets. The mixture was boiled at 95 °C for 8 min. Samples were detected by Western blotting. 

### 4.11. Mitochondrial Isolation 

Mitochondrial Isolation was performed using the Mitochondrial Isolation Kit (Invent Biotechnologies, Plymouth, MN, USA) following the manufacturer’s protocol. Briefly, cells (4 × 10^7^) were harvested at 600 g for 5 min. Cells were washed in cold PBS, resuspended in 250 μL buffer A, incubated on ice for 10 min and vibrated vigorously for 30 s. Cell suspension was transferred to a filter cartridge and centrifuged at 16,000× *g* for 30 s. The filter was discarded, and the pellet was resuspended and centrifuged at 3000 rpm for 1 min. The supernatant was mixed with 400 μL buffer B and then centrifuged at 16,000× *g* for 10 min. The supernatant was cytoplasmic protein. The pellet was resuspended in 200 μL buffer B and centrifuged at 7800× *g* for 5 min. The supernatant was mixed with 1.6 mL cold PBS and centrifuged at 16,000× *g* for 15 min. The pellet was isolated mitochondrial protein.

### 4.12. Mouse Experiments 

Experiments involving mice were performed in accordance with the National Institutes of Health Guide for the Care and Use of Laboratory Animals. Eight-week-old female C57BL/6 mice were obtained from Beijing Vital River Laboratory Animal Technology (Beijing, China). ID8 cells (6 × 10^6^), suspended in 100 μL PBS, were subcutaneously injected into the upper part of the mice groin. Two weeks later, mice were randomly divided into four groups (six mice per group) and given intraperitoneal injections of 1 mg/kg FCCP and/or 2 mg/kg cisplatin every 2 days. Two weeks later, mice were sacrificed, and tumors were dissected, weighed, and photographed. 

### 4.13. TUNEL Assay 

Cell apoptosis in tumor specimens was evaluated using the One Step TUNEL Apoptosis Assay Kit (Beyotime Biotechnology, Shanghai, China) according to the manufacturer’s instructions. Briefly, the sections of tumor tissue were dewaxed with xylene twice for 15 min, hydrated using an ethanol gradient (100% for 5 min, then 90% for 2 min, and 70% for 2 min), washed in distilled water for 2 min, then incubated with proteinase K at 37 °C for 30 min. The sections were washed with PBS three times and then incubated with TUNEL detection solution containing 25 μL terminal deoxynucleotidyl transferase and 225 μL fluorescence-labeled solution for 60 min at 37 °C, protected from light. After washing with PBS three times, the sections were counterstained with DAPI. The images were acquired using an Echo Lab Revolve microscope (San Diego, CA, USA).

### 4.14. Statistical Analysis

All results are based on at least three independent experiments. All data are given as mean values ± standard deviation (SD). Results between the two groups were compared using Student’s *t* test. Multiple comparisons were performed using ANOVA followed by Tukey posttest, *p* < 0.05 was considered statistically significant. Statistical analysis was carried out using GraphPad Prism 8.0 (La Jolla, CA, USA).

## Figures and Tables

**Figure 1 ijms-23-01320-f001:**
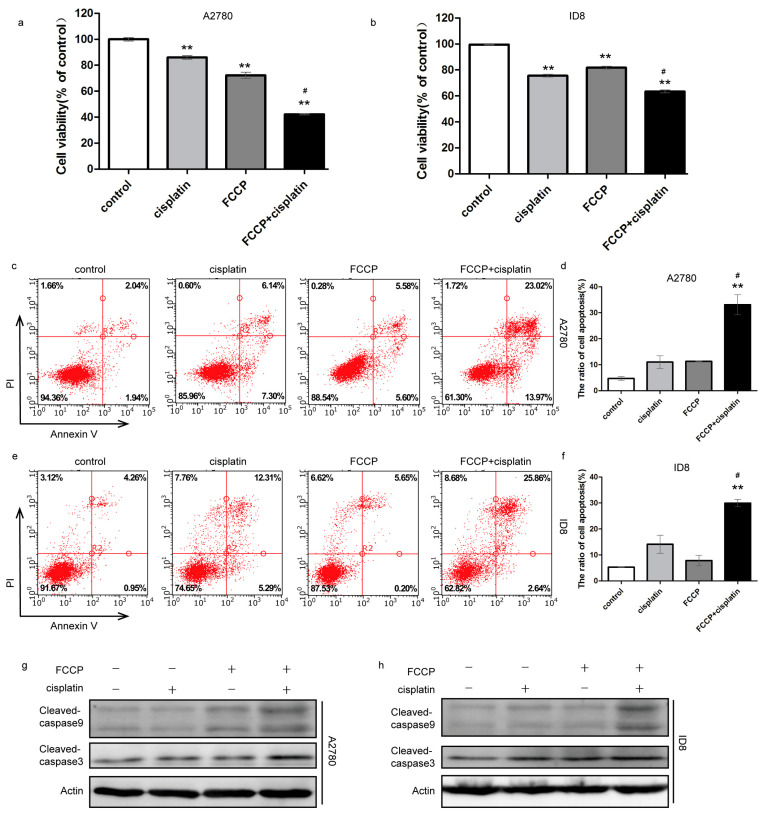
FCCP enhanced cisplatin-induced apoptosis in ovarian cancer cells. A2780 cells were treated with FCCP (2.5 μM) and/or cisplatin (1 μg/mL) for 24 h. ID8 cells were treated with FCCP (5 μM) and/or cisplatin (4 μg/mL) for 24 h. Cell viabilities were detected by MTT assay (**a**,**b**). Apoptosis rates were analyzed by flow cytometry (**c**–**f**). Data are presented as mean ± SD, *n* = 3. ** *p* < 0.01 vs. con, # *p* < 0.05 vs. cisplatin and vs. FCCP. Apoptosis-related protein levels were measured by Western blotting (**g**,**h**).

**Figure 2 ijms-23-01320-f002:**
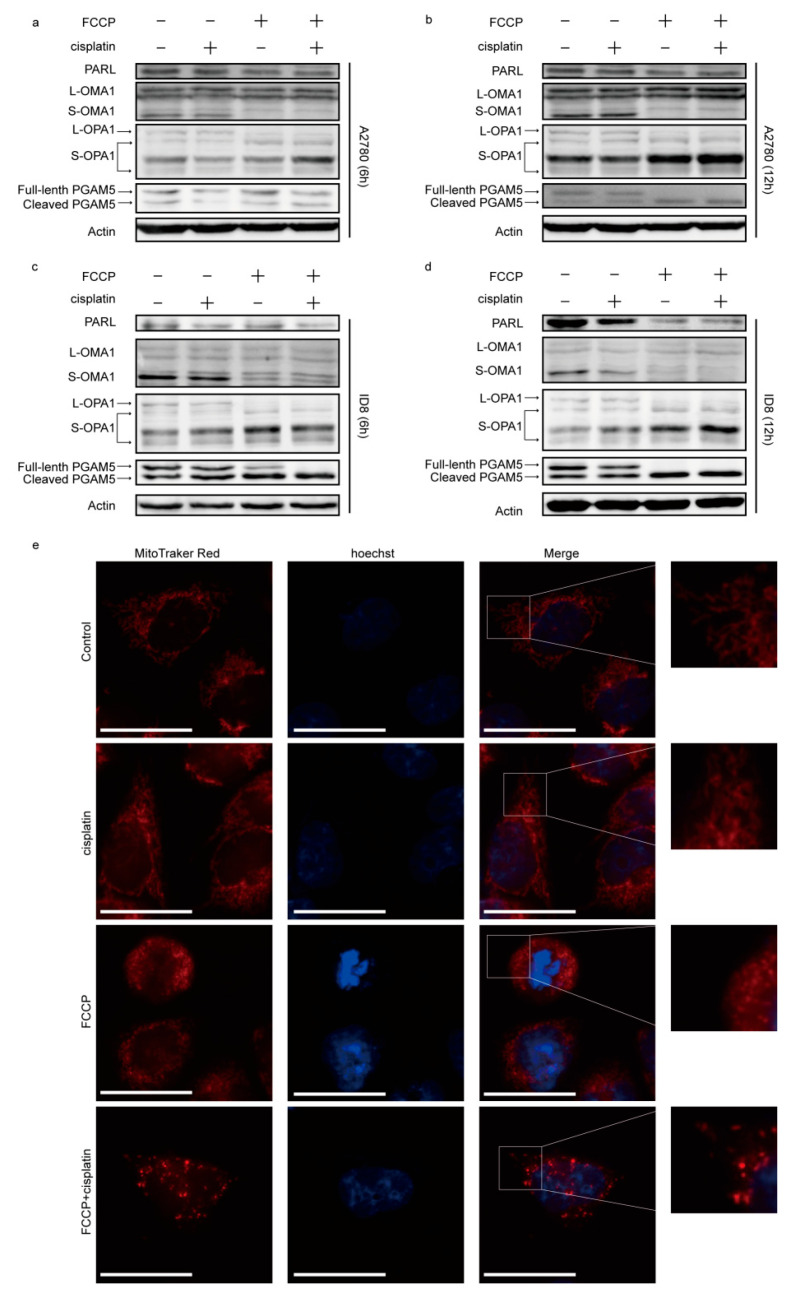
FCCP combined with cisplatin activated OMA1 and induced optic atrophy 1 (OPA1) cleavage and mitochondrial fragmentation. A2780 cells were treated with FCCP (2.5 μM) and/or cisplatin (1 μg/mL) for 6 and 12 h. ID8 cells were treated with FCCP (5 μM) and/or cisplatin (4 μg/mL) for 6 and 12 h. The expression of PARL, OMA1, OPA1, and PGAM5 was measured by Western blotting (**a**–**d**). A2780 cells were treated with FCCP (2.5 μM) and/or cisplatin (1 μg/mL) for 12 h, stained with Hoechst 33342 and MitoTracker Red and observed under a fluorescence microscope (scale bar, 30 μm) (**e**).

**Figure 3 ijms-23-01320-f003:**
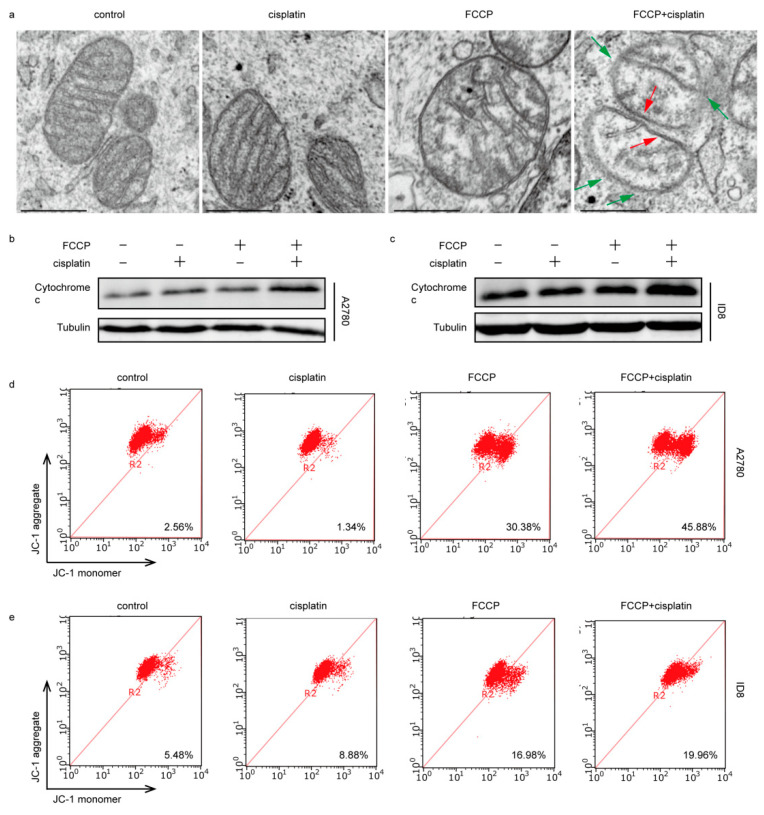
FCCP combined with cisplatin induced mitochondrial inner membrane (MIM) cristae remodeling and cytochrome c release. A2780 cells were treated with FCCP (2.5 μM) and/or cisplatin (1 μg/mL) for 12 h and observed under transmission electron microscopy. Red arrowheads indicated intact mitochondrial membranes. Green arrowheads indicated damaged mitochondrial membranes (scale bar, 0.5 μm) (**a**). A2780 cells were treated with FCCP (2.5 μM) and/or cisplatin (1 μg/mL) for 12 h. ID8 cells were treated with FCCP (5 μM) and/or cisplatin (4 μg/mL) for 12 h. The expression of cytochrome c in cytoplasm was measured by Western blotting (**b**,**c**). Mitochondrial membrane potentials were measured by JC-1 (**d**,**e**).

**Figure 4 ijms-23-01320-f004:**
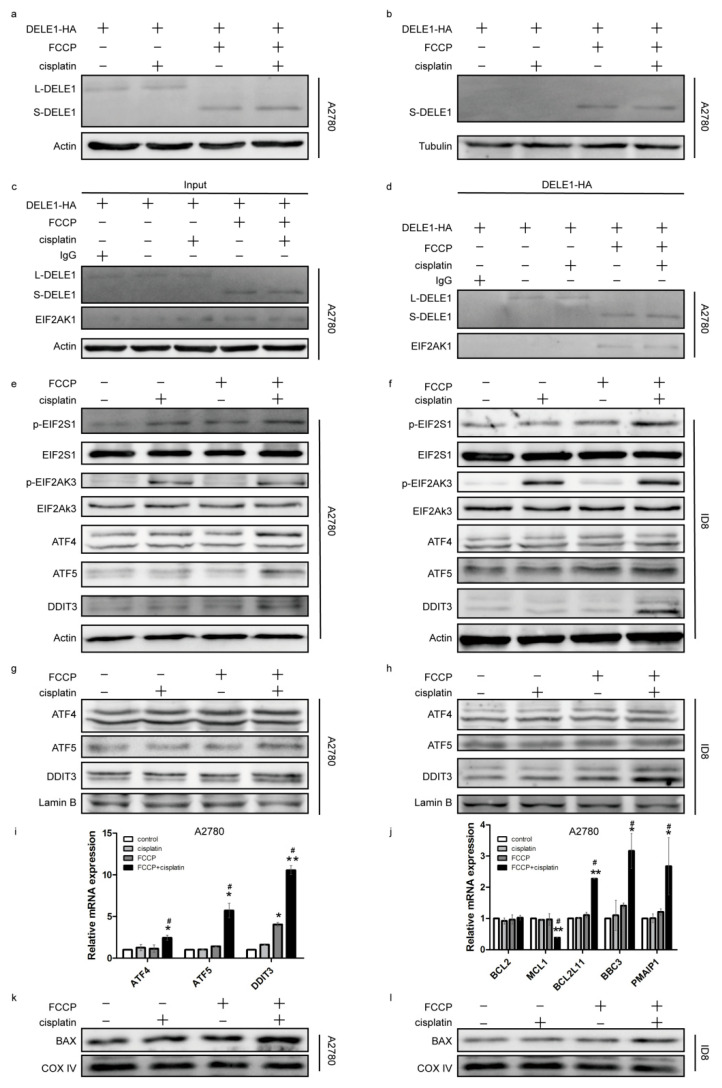
The activated OMA1 cooperated with EIF2AK3 to activate the EIF2S1/ATF4 pathway by cleaving DELE1. A2780 cells were transfected with DELE1-HA for 24 h, then treated with FCCP (2.5 μM) and/or cisplatin (1 μg/mL) for 6 h. The expression of DELE1 in whole cells (**a**) and in cytoplasm (**b**) was measured by Western blotting. Immunoprecipitations were performed with the anti-HA antibody followed by Western blotting using anti-EIF2AK1 antibody (**c**,**d**). A2780 cells were treated with FCCP (2.5 μM) and/or cisplatin (1 μg/mL) for 6 h. ID8 cells were treated with FCCP (5 μM) and/or cisplatin (4 μg/mL) for 6 h. The expression of p-EIF2S1, EIF2S1, p-EIF2AK3, EIF2AK3, ATF4, ATF5, and DDIT3 in the whole cell (**e**,**f**), and ATF4, ATF5, and DDIT3 in the nucleus (**g**,**h**) was measured. Relative mRNA expression of *ATF4*, *ATF5*, and *DDIT3* (**i**), and *BCL2*, *MCL1*, *BCL2L11*, *BBC3,* and *PMAIP1* (**j**) in A2780 cells was measured by qRT-PCR. Data are presented as mean ± SD, *n* = 3. * *p* < 0.05 vs. con, ** *p* < 0.01 vs. con, # *p* < 0.05 vs. cisplatin and vs. FCCP. A2780 cells were treated with FCCP (2.5 μM) and/or cisplatin (1 μg/mL) for 12 h. ID8 cells were treated with FCCP (5 μM) and/or cisplatin (4 μg/mL) for 12 h. The expression of BAX in the mitochondria was measured (**k**,**l**).

**Figure 5 ijms-23-01320-f005:**
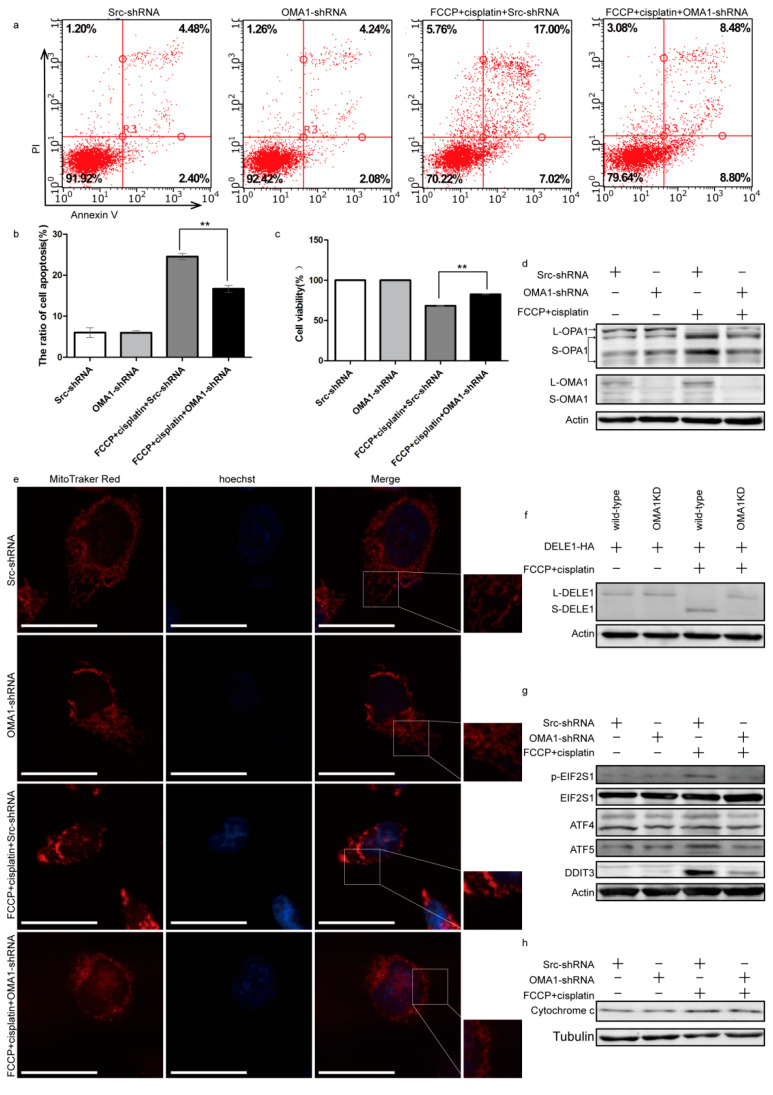
Knockdown of OMA1 partially reversed apoptosis by reducing the mitochondrial stress and the EIF2S1/ATF4 pathway. A2780 cells were transfected with OMA1-shRNA plasmids and Scr-shRNA for 24 h. After treatment with FCCP (2.5 μM) and/or cisplatin (1 μg/mL) for 24 h, apoptosis rates (**a,b**) and cell viability (**c**) were measured. Data are presented as mean ± SD, *n* = 3. ** *p* < 0.01 vs. FCCP + cisplatin + Src-shRNA. After treatment for 12 h, the expression of OMA1 and OPA1 (**d**) in the whole cell and cytochrome c in the cytoplasm (**h**) was evaluated. After treatment for 12 h, A2780 cells were stained with Hoechst 33342 and MitoTracker Red and observed under a fluorescence microscope (scale bar, 30 μm) (**e**). The wild-type A2780 cells and the OMA1 stable knockdown A2780 cells were transfected with DELE1-HA for 24 h, then treated with FCCP (2.5 μM) and/or cisplatin (1 μg/mL) for 6 h. Expression of DELE1 was evaluated (**f**). A2780 cells were transfected with OMA1-shRNA plasmids and Scr-shRNA for 24 h. After treatment with FCCP (2.5 μM) and/or cisplatin (1 μg/mL) for 6 h, the expression of p-EIF2S1, EIF2S1, ATF4, ATF5, and DDIT3 was evaluated (**g**).

**Figure 6 ijms-23-01320-f006:**
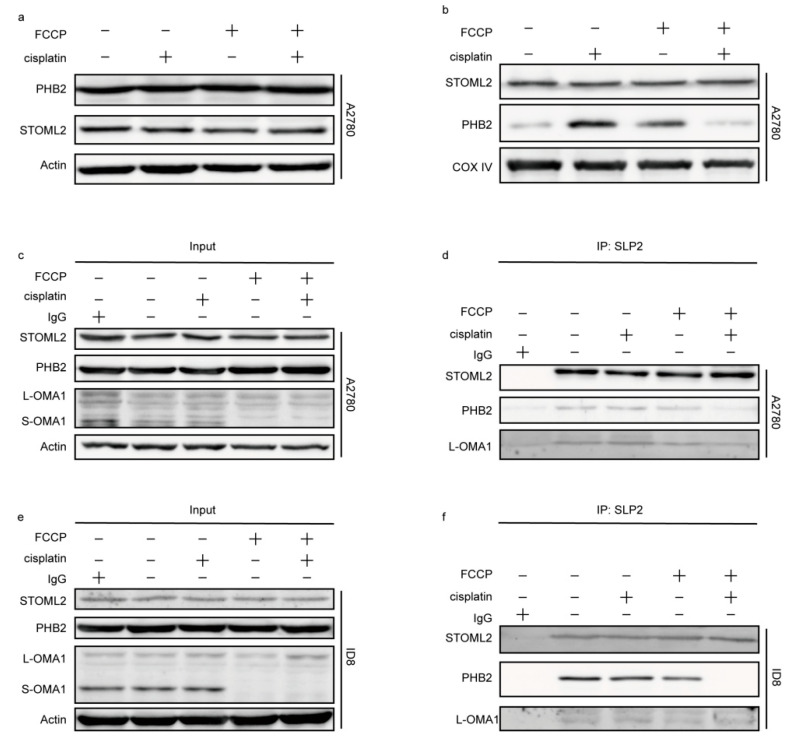
OMA1 protease activity was regulated by the prohibitin 2 (PHB2)/stomatin-like protein 2 (STOML2) complex. A2780 cells were treated with FCCP (2.5 μM) and/or cisplatin (1 μg/mL) for 12 h. Expression of PHB2 and STOML2 in the whole cell (**a**) and in the mitochondria (**b**) was measured. A2780 cells were treated with FCCP (2.5 μM) and/or cisplatin (1 μg/mL) for 12 h. ID8 cells were treated with FCCP (5 μM) and/or cisplatin (4 μg/mL) for 12 h. Immunoprecipitations were performed with the anti-STOML2 antibody followed by Western blotting using anti-PHB2 and anti-OMA1 antibodies (**c**–**f**).

**Figure 7 ijms-23-01320-f007:**
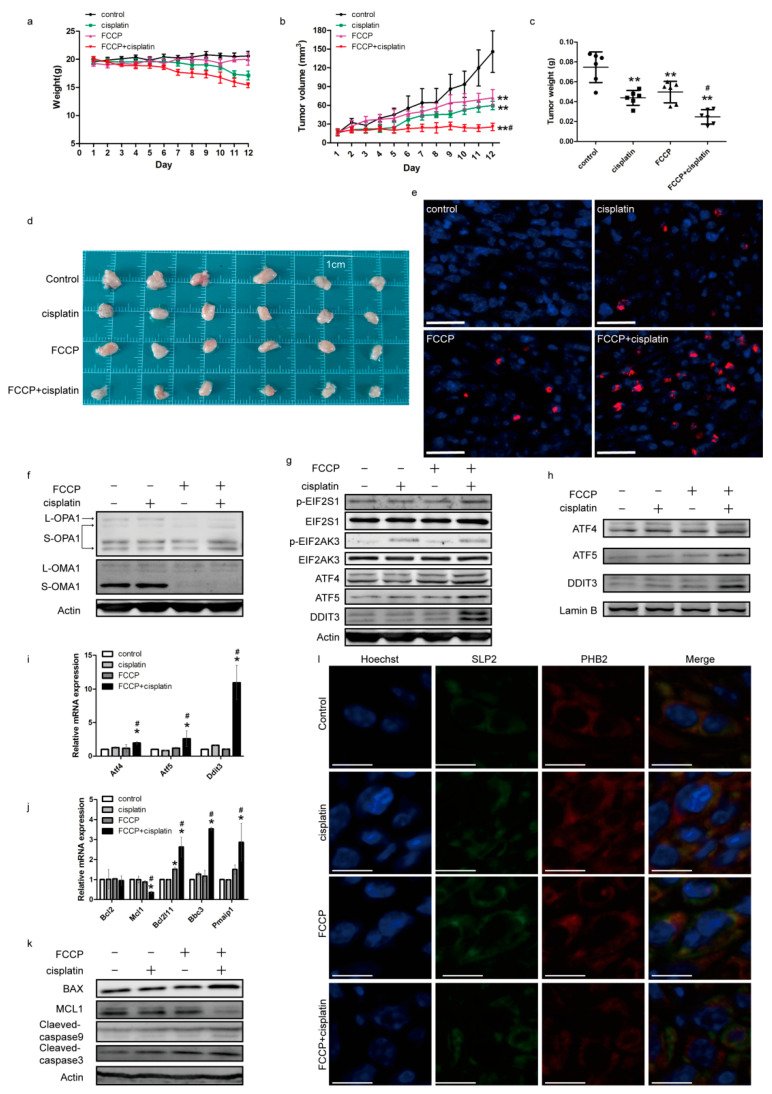
Effects of FCCP and cisplatin treatments in a mouse subcutaneous tumor model. ID8 cells were subcutaneously implanted into C57BL/6 mice. Mice were treated with 1 mg/kg FCCP and/or 2 mg/kg cisplatin for 12 days (each group, *n* = 6). Body weight and tumor volume were measured daily (**a**,**b**). The wet weight of tumors was measured at autopsy (**c**). Images of excised tumors (**d**). Representative images of TUNEL assay from subcutaneous tumors (scale bar, 30 μm) (**e**). Red fluorescence indicates TUNEL-positive cells. Subcutaneous tumors were lysed in RIPA and the expression of OPA1, OMA1 (**f**), p-EIF2S1, EIF2S1, p-EIF2AK3, EIF2AK3, ATF4, ATF5, DDIT3 (**g**), ATF4, ATF5, DDIT3 in the nucleus (**h**), BAX, MCL1, cleaved caspase-3, and cleaved caspase-9 (**k**) was measured. Relative mRNA expression of *Atf4*, *Atf5*, and *Ddit3* (**i**), and *Bcl2, Mcl1*, *Bcl2l11*, *Bbc3*, and *Pmaip1* (**j**) was measured. Data are presented as mean ± SD, *n* = 3. * *p* < 0.05 vs. con, ** *p* < 0.01 vs. con, # *p* < 0.05 vs. cisplatin and vs. FCCP. Localization of STOML2 and PHB2 in tumor specimens was measured by immunofluorescence staining (scale bar, 10 μm) (**l**).

**Figure 8 ijms-23-01320-f008:**
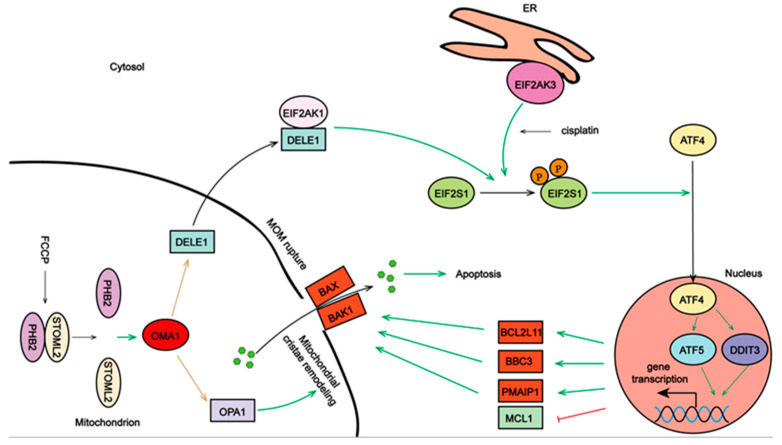
The mitochondrial PHB2/OMA1/DELE1 pathway cooperates with endoplasmic reticulum stress to induce ovarian cancer cell death. Destruction of the prohibitin 2 (PHB2)/stomatin-like protein 2 (STOML2) complex releases OMA1 protease activity. Activated OMA1 cleaves optic atrophy 1 (OPA1) and DELE1, resulting in mitochondrial cristae remodeling and DELE1’s cytoplasmic interaction with EIF2AK1. Activated EIF2AK1 cooperates with EIF2AK3 to induce EIF2S1 phosphorylation, promoting the transfer of ATF4 into the nucleus and the upregulation of DDIT3 and ATF5. Then, pro-apoptotic proteins, BBC3, BCL2L11 and PMAIP1, are upregulated and anti-apoptotic protein MCL1 is downregulated, triggering accumulation of BAX and BAK1 on the mitochondrial outer membrane (MOM) and the rupture of the MOM. Under the coordination of mitochondrial inner and outer membranes, cytochrome c is released, and the cell dies. Green arrows indicate positive regulation; red arrow indicates negative regulation.

**Table 1 ijms-23-01320-t001:** Primer sequences.

Primer Name Human	Sequence	Primer Name Mouse	Sequence
*ATF4*	5′-GTTTTGGATTGGTGGGGTGC-3′ 5′-TCTTGGTTCCTGCCACGTTT-3′	*Atf4*	5′-GCCGGTTTAAGTTGTGTGCT-3′ 5′-CTGGATTCGAGGAATGTGCT-3′
*ATF5*	5′-TCAGGAGGAGGAAACCAGACC-3′ 5′-CCTCGGCTGAAGAAAGAAGGT-3′	*Atf5*	5′-GGCTGGCTCGTAGACTATGG-3′ 5′-CCAGAGGAAGGAGAGCTGTG-3′
*DDIT3*	5′-GGAAACAGAGTGGTCATTCCC-3′ 5′-CTGCTTGAGCCGTTCATTCTC-3′	*Ddit3*	5′-CCTAGCTTGGCTGACAGAGG-3′ 5′-CTGCTCCTTCTCCTTCATGC-3′
*PMAIP1*	5′-CACGAGGAACAAGTGCAAGTAG-3′ 5′-TGATGAAACGTGCACCTCCT-3′	*Pmaip1*	5′-GACAAAGTGAATTTACGGCAGA-3′ 5′-GGTTTCACGTTATCACAGCTCA-3′
*BCL2L11*	5′-TAAGTTCTGAGTGTGACCGAGA-3′ 5′-GCTCTGTCTGTAGGGAGGTAGG-3′	*Bcl2l11*	5′-ACTTGGATTCACACCACCTCC-3′ 5′-GTCGGGATTACCTTGCGGTT-3′
*BBC3*	5′-GAGCAGGGCAGGAAGTAACAA-3′ 5′-GAGGGCTGAGGACCACAAAT-3′	*Bbc3*	5′-TGTTTGGCTCCCGAGTTTGT-3′ 5′-CCAGGCAAGCGACAGATACA-3′
*MCL1*	5′-TGCTTCGGAAACTGGACATCA-3′ 5′-TAGCCACAAAGGCACCAAAAG-3′	*Mcl1*	5′-CAAACTGGGGCAGGATTGTG-3′ 5′-CACATTTCTGATGCCGCCTT-3′
*BCL2*	5′-TTCTTTGAGTTCGGTGGGGT-3′ 5′-GCCCATGCTGAAACTCCCTTA-3′	*Bcl2*	5′-CGTCGTGACTTCGCAGAGAT-3′ 5′-AGTTCCACAAAGGCATCCCA-3′
*ACTB*	5′-TTTCTGAGTTGATTTCCCGGTC-3′ 5′-ACCGAACTTGCATTGATTCCAG-3′	*Actb*	5′-GTATGGAATCCTGTGGCATC-3′ 5′-AAGCACTTGCGGTGCACGAT-3′

## Data Availability

Data sharing not applicable.

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
