# Peer review of "The Mitochondrial PHB2/OMA1/DELE1 Pathway Cooperates with Endoplasmic Reticulum Stress to Facilitate the Response to Chemotherapeutics in Ovarian Cancer"

_ijms, 2022, doi:10.3390/ijms23031320_

Round 1

Reviewer 1 Report

Cheng and colleagues presented an interesting research article aimed at evaluating the association between mitochondrial pathways and endoplasmic reticulum stress in the response to cisplatin in ovarian cancer. For this purpose, the authors performed several in vitro and functional experiments on ovarian cancer cell lines in order to identify the mitochondrial molecular determinants associated with cisplatin response as well as to unveil alteration in the cellular structure after anticancer treatments. Overall, the study is well-conceived and the experimental design appears complete and detailed. However, there are some aspects that the authors have to clarify before publication. Please see the comments below:
1) In the first part of the Introduction section, the authors have to better emphasize the impact of cisplatin resistance in ovarian cancer and thus the importance of studying in a multidisciplinary manner the molecular mechanisms responsible for the efficacy or resistance of ovarian cancer to these drugs. For this purpose, please see:
- PMID: 34132354
- PMID: 32269724
- PMID: 31450627
- PMID: 33076245
2) In the following sentence, it should be “PARL”, please check: “...RARL was reduced, and OPA1 and PGAM5 were cleaved in the group treated with FCCP  alone...”; 
3) In chapter 2.2, please clarify why only the protein levels of OMA1, OPA1 and PARL without evaluating the mRNA expression;
It is not clear why the authors perform some experiments only on A2780 cells and others only on ID8 cells. This could represent a limitation of the study. Please, clarify;
4) In the Methods section, please indicate how many cells were seeded for each experiment (e.g. cell transfection, cell viability assay, etc.);
5) In chapter 4.6, please indicate the dilutions of the antibodies used;
6) Please be more detailed in the description of the methods used in chapters 4.11 and 4.13.

Reviewer 2 Report

In the submitted manuscript Cheng et al. inspected the role of mitochondrial PHB2/OMA1/DELE1 pathway in the response of ovarian cancer cell lines to chemotherapeutics. They have shown that activation of OMA1 increased ovarian cancer cells' sensitivity to cisplatin in vivo and in vitro through OPA1 cleaving, what leads to mitochondrial inner membrane cristae remodeling.  Simultaneously, OMA1 induced DELE1 cleavage and its cytoplasmic interaction with HRI. They have also demonstrated that HRI cooperated with the endoplasmic reticulum stress sensor PERK to amplify elF2α/ATF4 signal, what resulted in the rupture of the mitochondrial outer membrane. Furthermore, they have shown that OMA1 knockdown attenuated all these activities and reversed apoptosis. Additionally, they have found that OMA1 protease activity was regulated by the prohibitin 2 (PHB2)/Stomatin-like protein 2 (SLP2) complex.

The manuscript is well written, English quality is high, enough number of adequate experiments were conducted, and conclusions were corroborated by the results.

However, there are several things which must be additionally improved.

1) State in the text that A2780 are human endometrioid ovarian adenocarcinoma cells and that ID8 are mice epithelial ovarian cancer cells, so that it would be clear that your results cannot be a priori pertained to all sub-types of ovarian cancer.

2) Since majority of your results came from western blotting, blots presented on figures must be quantified with a software like ImageJ and actual statistical analysis performed on protein expression levels, especially since you have several blots with pale bands and very dark background, and you also use word "significantly" to describe comparison of protein expression levels.

3) Several methods need additional details:

  • for all cell-based assays, provide number of plated cells
  • provide dilutions for ALL used antibodies
  • since in the reference [25] this information is missing, provide qPCR cycling conditions, state again which gene was used for normalization, and if you have use 2^-ddCt method to calculate relative gene expression provide its reference (doi: 10.1006/meth.2001.1262)
  • provide model and manufacturer of microplate reader used in "4.4 Cell Viability Assay"
  • in "4.14 Statistical analysis" subsection and figure legends correct statement because all your bar graphs present only mean + SD, not ±; provide which post-hoc test was used with ANOVA

4) For the sake of unambiguity and standardization, pleas refer to https://www.genenames.org/ and https://www.uniprot.org/uniprot/ and use only approved and recommended names and symbols of human genes and protein. E.g., Diablo is recommended name for Smac protein, proper symbols for human Mcl-1 and Bcl-2 genes (Table 1) are MCL1 and BCL2, Greek letters should not be used (eIF-2A). Also keep in mind that style for writing human and mice gene and protein symbols differs (https://www.biosciencewriters.com/Guidelines-for-Formatting-Gene-and-Protein-Names.aspx).

5) In figure legends provide compound concentrations used for treatments. Provide explanation for "con" abbreviation, or rather put whole word "control" on figures.

6) Line 410: I suppose the correct sentence would be "RNA extraction and qRT-PCR were performed as described before [25]."

Reviewer 3 Report

In this interesting manuscript, using in vitro and in vivo experimental work, the authors concluded that mitochondrial PHB2/OMA1/DELE1 pathway cooperates with ER stress to facilitate the response to chemotherapy in ovarian cancer. 

There are several issues related to the results and methods, which should be addressed and fixed first:

1-In figure 3 A, showing the TEM of the combined treatment group, the arrows mark mitophagosomes not damaged mitochondria. Please provide an image showing the whole mitochondrion will clear outer and inner mitochomdrial membranes

2-Regarding apoptotic cells shown by TUNEL method, please show these apoptosis cells by TEM.

3-Please describe the details of one step TUNEL assay

4-It will be great if you provide additional figure showing a a schema of your results and conclusions

5-Regarding PHB2, Why it is downregulated in the combined treatment group?

Round 2

Reviewer 3 Report

The manuscript is improved and the authors responded to my comments

Thank you